# Work engagement, impulsivity and, self-efficacy among Polish workers. Moderating role of impulsivity

**Bohdan Rożnowski**[1]◉*, **Antoni Wontorczyk**[2]◉

**1** Faculty of Social Sciences, Institute of Psychology, John Paul II Catholic University of Lublin, Lublin, Poland, **2** Faculty of Management and Social Communication, Institute of Applied Psychology, Jagiellonian University, Kraków, Poland

◉ These authors contributed equally to this work.
* bohdan.roznowski@kul.pl

**Data Availability Statement:** The dataset called Replication Data for "ImpulsivityWEngagement" (view from https://uj.rodbuk.pl/dataset.xhtml?persistentId=doi:10.57903/UJ/I7WTSH) has been created in Faculty of Management and Social

## Abstract

The study proposes adding a new component to the Job Demands-Resources Theory, termed 'personal demands'. Impulsivity serves as an illustrative example of personal demands. A cross-sectional questionnaire was conducted and a total of 394 (Female = 59.6%) workers were surveyed. Standardized survey questionnaires were used in the study, i.e. The Utrecht Work Engagement Scale (UWES-9), Life Roles Self-Efficacy Scale (LRSES) and UPPS-P Impulsive Behaviour Scale. The obtained results indicate that Work Engagement correlates negatively with Impulsivity scales (lack of premeditation ($r = -.22$; $p < .001$), lack of persistence ($r = -.27$; $p < .001$), positive urgency ($r = -.12$; $p, < .01$) and negative urgency ($r = -.12$; $p < .01$)). In addition, Positive Urgency, moderates the relation between Self-efficacy and Work Engagement, in line with the theory ($B = .133$; $SE = .067$; $t = 1.98$; $p < .05$; $LLCI = .0007$; $ULCI = 0.2643$). This confirmed the fact, that impulsivity should be included in the JDR theory as a 'personal demand'.

## Introduction

Since Shuck and Wollard [1] published their study of employee work engagement, a steady increase has been observed in researcher interest in the issue. Work engagement has been investigated in different domains and with regard to various consequences in the organization. In recent years researchers found that work engagement is a mediator of goal attainment and remote work [2], job outcomes [3] turnover intention [4], affective organizational commitment [5], innovative performance [6,7], employee creativity [8] and job crafting [9].

The concept of work engagement appeared in subject related literature as early as Kahn's [10] work at the end of the 20th century. Kahn conceptualized engagement as an employee motivational mechanism. He argued that three psychological conditions need to be met in order for employees to become engaged: meaningfulness (i.e. the sense of receiving a gratification from 'investing' oneself in a role), psychological security (i.e. the feeling of being able to prove oneself and assign oneself to a job without fear of negative consequences) and

Communication (view from https://uj.rodbuk.pl/dataverse/wzks).

**Funding:** This research received no external funding.

**Competing interests:** The authors have declared that no competing interests exist.

availability (i.e. the belief that one has the physical and psychological resources of work engagement). Work engagement refers to a rational choice, an individual's decision-making about the degree to which he or she will devote himself or herself to the performance of a work role [11]. However, work engagement is most commonly understood as the positive motivational state of an employee expressing high vigor, absorption and dedication [12,13]. It is a state that promotes better job performance, taking initiatives to improve the work situation—job crafting [14,15].

In this article, we take the definition proposed by Bakker at al. [16] as the motivational state for the basis of analysis. Generally speaking, this is a temporary condition, resulting from a temporary constellation of labour resources. This is emphasized by Schaufeli and DeWitte [17] who wrote that engagement changes not only from day to day, but also from task to task. An employee may be engaged in one task (e.g., giving advice to a client) on the same day and not feel that way on the next (administratively documenting the advice given). Therefore, stable sustainability of resources in a work situation is of key importance. It should be taken into account that the importance of individual resources is relative and varies from worker to worker [18]. Bakker and de Vries [19] proposed the distinction of stable resources, which they divided into organizational (eg. Human Resources Practices, Leadership etc.) and key personal resources e.g. Proactive personality, Self-Efficacy, Emotional Intelligence, Conscientiousness) (p. 6). Stable resources provide a guarantee of relatively sustained work engagement. Hence, it is worthwhile to undertake the analysis of such stable key personal resources in research.

Currently, the job demands-resources (JD-R) theory is the dominant one while planning and implementing research in the area of organisational behaviour. The JD-R model was introduced in subject-related literature years ago [20]. The concept has gradually evolved from a detailed model explaining work engagement or job burnout [3] towards a complex theory [14,15]. In the first version of the Job Demands—Job Resources Model, only two categories of job characteristics were included: job demands and job resources. The following definitions have been provided to better understand the two categories—"Job Demands are defined as those physical, psychological, social, or organizational aspects of the job that require sustained physical and/or psychological effort and are therefore associated with certain physiological and/or psychological costs" [20] (p. 501) and "Job Resources refer to those physical, psychological, social, or organizational aspects of the job that are functional in achieving work goals, reduce job demands and the associated physiological and psychological costs, or stimulate personal growth, learning, and development" [20] (p. 501). As numerous studies were conducted, the model was expanded to include an additional category of Personal Resources—personal characteristics that help individuals cope with the demands of work [21]. These are defined as: 'Personal resources refer to the beliefs people hold regarding the extent of control they have over their environment.' [14] (p.275). Later, Demerouti and Bakker [14] noted that there may still be personal factors hindering work–referred to as "Personal Demands". As suggested by Barbier and colleagues [22], personal demands can be defined as "the requirements that individuals set for their own performance and behavior that force them to invest effort in their work and are therefore associated with physical and psychological costs" (p. 2). The main relationship described by JD-R theory indicates that resources including both work resources and personal resources stimulate work motivation. It manifests itself in greater work engagement, commitment and flourishing [14] (p 275). On the other hand, Bakker and Demerouti [3] identify a group of factors that could be called 'job demands'. These factors contribute to negative work experiences and translate into negative aspects of work.

The authors of the JD-R theory mentioned here the following psychological states: Exhaustion, Job-related anxiety and Health complaints as the effects of work overload, emotional job demands, physical job demands, and work-home conflict [14]."The personal resources have a

direct positive effect on work engagement. In addition, personal resources are expected to buffer the undesirable impact of job demands on strain, and boost the desirable impact of (challenge) job demands on motivation" [14] (p 275).

To now, personal demands have received little attention in subject-related literature as well as in research conducted within the JD-R model. Bakker and Demrouti [14] believe that work-aholism is the most studied issue. At the same time, they point out that the personal demands aspect is an "intriguing topic". There is a need to identify additional indicators of personal demands, which are concepts well-established in psychology but currently absent from JD-R theory, such as temperamental variables like impulsivity. In psychological literature, impulsiv-ity is most often understood as a factor that impedes personal and social functioning [31,33,34,40,43,62]. We assume, therefore, that it can be one of the important factors contrib-uting to job demands that arise from personal characteristics.

In the present study, we attempted to investigate if there are any specific characteristics linked to the dispositional features of the employee, which have been overlooked up to now in the research conducted within the JD-R model, which makes it possible to complement a pre-viously missing aspect–personal demands. Of particular interest is the issue of the negative impact of personal demands on the aspect of motivation (work engagement) distinguished in the theory. Therefore, we intended to analyse the impact of a personal construct genetically determined by temperamental traits–impulsivity on the employee's motivational states of behaviour.

## Self-efficacy

In JDR theory, self-efficacy is very often mentioned among personal resources strongly related to work engagement[23–25]. All studies consistently indicate a positive association of self-effi-cacy attitude in the work domain with engagement and other positive phenomena, e.g. job sat-isfaction, employee well-being.

The term self-efficacy is derived from A. Bandura [26], who presented a broad perspective of the determinants and consequences of an individual having self-efficacy beliefs. These include stronger motivation to achieve goals, greater commitment to achieving goals, resis-tance to discouragement and greater perseverance, which together result in the greater effec-tiveness of actions taken.

There are many synonymous terms in subject-related literature to describe the phenome-non of self-efficacy: professional, vocational, job, work, career, etc. All terms describe the same phenomenon—the subject's belief that he or she can cope with the demands of the job and with the role assigned to him or her at work [27]. A strong belief in self-efficacy is a buffer for job burnout [28], a predictor of work ability [29], while also a factor influencing the reduction of mistakes made at work [30].

## Impulsivity

An employee's personal resources, which are relatively stable over his/her lifetime, are the per-sonal traits, in particular, impulsivity. Impulsivity is a complex construct variously defined in psychology [31,32]. It is most typically understood as a predisposition to quick, unplanned responses to internal or external stimuli with limited consideration of the negative conse-quences of one's behaviour [33], the lack of control over behaviour [34–39], poor emotional regulation [40–42] and also the inability to postpone gratification [43–46]. The impulsive behaviour of individuals leads to a number of negative consequences: sensation seeking in adults and adolescents [47–49], drug-taking [50,51]; eating and diet use disorders [52–56], alcohol and other substance abuse [51,57,58]; pathological gambling [49,59,60]; risky sexual

behaviour [61]. Impulsive behaviour is associated with the lack of attention, lack of planning and taking immediate action. Studies have identified impulsive personality as a socially maladaptive (even psychopathological) trait [62]. Therefore, it can be assumed that impulsive behaviour, as well as impulsive personality, will also be associated with a person's work activity, particularly work engagement.

A similar view of impulsivity as a personal trait leading to negative behavioural consequences is also presented by [63]. In our study, impulsivity as understood by Whiteside and Lynam was included as a personal disposition in the area of Demands in the JD-R model [14,15]. It was assumed that, as in the case of Resources where the authors of the JD-R model distinguished between Job Resource and Personal Resources, similarly to the case of Demands, it is possible to distinguish between Job Demands and Personal Demands. Barbier et al.[22] defined personal demands as "the requirements that individuals set for their own performance and behaviour that force them to invest effort and are therefore associated with physical and psychological costs" [22] (p.762).

Assuming this understanding of Personal Demands and the fact that impulsivity leads to negative behavioural consequences–as has been shown in a number of studies [46,64,65]–it was determined that impulsivity would also have negative consequences for behaviour at work. The main aim of the study was to describe the relationship between Impulsivity and Work Engagement. We consider both the direct relationship in which impulsivity is a predictor of engagement and the indirect relationship in which impulsivity is the moderator of the relationship between self-efficacy beliefs and engagement.

Therefore, we pose the research question of whether the construct Impulsivity, included in the study of work engagement according to the JD-R theory, will meet the conditions for being considered a personal requirement and two hypotheses were formulated as follows:

H1. Impulsivity as the dispositional characteristic of an employee is a negative predictor of work engagement.

According to the J-D R model, resources stimulate work engagement while at the same time, demands increase the level of negative work experiences. Negative work experiences reduce the level of work engagement of employees. Thus, it can be assumed that dispositional traits (impulsivity), when required to put more effort into doing the job, will negatively affect engagement [3]. Impulsivity that hampers an employee's social functioning will constitute Personal Demands (analogous to the dichotomous distribution of resources) and, according to the J-DR model, these will negatively impact the work engagement level.

H2. Impulsivity as a dispositional trait of an employee is the moderator of the relationship between self-efficacy beliefs as an employee and work engagement.

As Bakker and Demeroouti [3] suggest, job demands can lead to lower work engagement. Similarly, impulsivity as personal demand will lead to lower job engagement. However, in a work situation, high levels of impulsivity in an employee will be a significant moderator reinforcing the association of self-efficacy with work engagement. Self-efficacy leading to greater work engagement will be additionally positively enhanced when impulsivity is high.

## Methods

### Research tools

The three crucial variables for the study (impulsivity, self-efficacy, and work engagement) were operationally defined (measured) using standardized, validated questionnaires which have been previously used in other studies.

To measure the variable work engagement, the Utrecht Work Engagement Scale by Schaufeli and Baker [13] was chosen as the most popular questionnaire for measuring this variable. We selected the 9-item version because it gets just as good rates as the 17-item version [66] and is shorter. The scale has been adapted into Polish [67]. The scale in our study had satisfactory alpha reliability indices (Vigor $\alpha$ = .81, Absorption $\alpha$ = .67; Dedication $\alpha$ = .77).

The variable belief in self-efficacy was measured by the questionnaire known as the Life roles self-efficacy scale [27]. This tool is based on the assumption of parallel human functioning in multiple social roles, which can be perceived as domains of efficacy. The questionnaire measures self-efficacy in five major social roles: Student, Worker, Homemaker, Leisurite and Citizen. In the paper the role of the Worker was the most important. The measurement is performed with 3 item sets relating to each role. The Respondent answers the statements on a 6-point scale. The questionnaire has been used in a number of studies and has proved to be relevant and reliable. In the study Crombach's $\alpha$ obtains: Student Self-efficacy $\alpha$ = .78, Worker Self-efficacy $\alpha$ = .71; Homemaker Self-efficacy $\alpha$ = .78; Leisurite Self-efficacy $\alpha$ = .75; Citizen Self-efficacy $\alpha$ = .83.

Impulsivity was measured by the SUPPS-P impulsive behaviour scale [42]. The tool is an abbreviated 20-item Likert-type self-report scale consisting of five subscales: Positive Urgency, Negative Urgency, Lack of Premeditation, Lack of Perseverance and Sensation Seeking. Sample questions for each scale are as follows: "When I am really ecstatic, I tend to get out of control; When I am upset, I often act without thinking; I like to stop and think things over before I do them; I generally like to see things through to the end; I would enjoy parachute jumping". Items are rated on a four-point scale from 1 (strongly agree) to 4 (strongly disagree). The UPPS-P has satisfactory psychometric properties in terms of both convergent and discriminant validity [16,18,43]. In our study, we used the Spanish version of the questionnaire, which was adapted into Polish [68].

## Procedure

The survey was conducted from 15th of November 2021 to the 31-st of December 2021. The results presented in this text are part of a larger research project on personal requirements and resources related to work engagement. The choice of the research timeframe was not influenced by external factors, but only by the realization of the empirical goal. Due to the SARS--COVID 19 pandemic, direct contact between interviewers and respondents was avoided. Trained interviewers contacted potential respondents by phone and invited them to participate in the research. At the beginning of the conversation, the purpose of the study and the conditions for participation were presented (including voluntariness, anonymity, and the ability to withdraw from the study at any stage). The person at hand was then asked to agree to participate in the study. Those who did not consent were thanked for their time and contact was terminated. Those who agreed to participate were sent links to the questionnaire in Google Forms with a repeat request to complete it. Since the questionnaire was shared with those who gave their consent during the phone call, it did not include a repeat question about consent. Despite declaring their consent to participate in the study, participants were free to drop out at any time during the study. Participants were also informed that withdrawal from the study during the course of the study would not result in any negative consequences. The procedure for conducting the study was approved by the Research Ethics Committee of the Institute of Applied Psychology of the Jagiellonian University (Appendix No. 109/1A/2021). The study was conducted in accordance with the recommendations of the Declaration of Helsinki.

Table 1. Sample demographics.

| Variable | M | SD | |
|---|---|---|---|
| Age | 30.57 | 9.64 | |
| Seniority | 5.02 | 6.18 | |
| Variable | Value | N | % |
| Sex | Female | 235 | 59.6 |
| | Male | 159 | 40.4 |
| Education | Primary | 6 | 1.5 |
| | Vocational | 25 | 6.3 |
| | Secondary | 154 | 39.1 |
| | University Degrees | 211 | 53,0 |
| Position | Manager | 72 | 18,3 |
| | Professional | 168 | 42,6 |
| | Worker | 154 | 39,1 |

## Sample

The survey targeted a working population living in three provinces in southern Poland: Malopolskie, Podkarpackie and Swietokrzyskie. We chose the residents of these three regions for the study because they have different historical and cultural traditions, which helped ensure the cultural diversity of the sample. The sample size was set at over 250 to meet the requirements for conducting multivariate regression analyses. The study included 394 subjects (59.6% Female). The average age was M = 30.6; SD = 9.64. The seniority of the respondents amounted to 5 years (SD = 6.20). More than half of the respondents had higher education (53%). Completion of secondary school education was indicated by 39% of the respondents. Most were single without children (38.3%) or living in a relationship without children (30.5%). More details are included in Table 1.

## Statistics

The data were statistically analyzed using the IBM SPSS Statistics package version 28 with the Hayes Process overlay. In describing the data, reference was made to means, standard deviations and Pearson's r- correlation coefficient, since the distribution of results met the conditions for parametric methods. In-depth analyses of the variables' relationships were made with regression equations using semi-particle correlations to eliminate distortions due to inter-correlation of explanatory variables. In order to test the moderation model, Model 1 was calculated using Hayes' Process [69] overlay and dedicated to mediation and moderation analyses, was used. In this analysis, in addition to the relationships of individual explanatory variables (self-efficacy and impulisivity) with the explained variable (work engagement), the effect of interaction between explanatory variables is also analyzed as a separate factor affecting the explained variable. Path Relevance Interaction means that specific combinations of values of explanatory variables have different relationships with the explained variable.

## Results

### Preliminary analyses of descriptive statistics and correlations

The results included in Table 2 indicate that the respondents demonstrated a level of intensity of work engagement only slightly higher than half of the possible points to be scored (M = 32.7; SD = 8.68). At the same time, they were characterized by a fairly high level of self-

**Table 2. Descriptives and correlations (N = 394).**

| No | Variables | M | SD | 1 | 2 | 3 | 4 | 5 | 6 | 7 | 8 | 9 | 10 |
|----|-----------|----|----|----|----|----|----|----|----|----|----|----|----|
| 1 | Work engagement | 32.78 | 8.69 | | | | | | | | | | |
| 2 | Lack of premeditation | 7.98 | 2.04 | -.220*** | | | | | | | | | |
| 3 | Lack of perseverance | 7.79 | 2.17 | -.274*** | .614*** | | | | | | | | |
| 4 | Negative urgency | 13.63 | 2.14 | -.124** | .354*** | .267*** | | | | | | | |
| 5 | Positive urgency | 13.15 | 2.17 | -.121** | .367*** | .196*** | .575*** | | | | | | |
| 6 | Sensation seeking | 14.83 | 2.47 | .052 | -.165*** | -.26***3 | -.004 | .156*** | | | | | |
| 7 | Self-efficacy student | 14.01 | 2.26 | .340*** | -.391*** | -.428*** | -.254*** | -.305*** | .161** | | | | |
| 8 | Self-efficacy worker | 13.46 | 2.29 | .446*** | -.430*** | -.399*** | -.294*** | -.265*** | .229*** | .711*** | | | |
| 9 | Self-efficacy homemaker | 14.17 | 2.41 | .305*** | -.346*** | -.410*** | -.129** | -.220*** | .087* | .728*** | .514*** | | |
| 10 | Self-efficacy leisurite | 12.24 | 2.72 | .178*** | -.115** | -.128** | -.146** | -.059 | .110** | .392*** | .483*** | .285*** | |
| 11 | Self-efficacy citizen | 12.31 | 2.59 | .238*** | -.136** | -.172*** | -.115** | -.041 | .119** | .305*** | .326*** | .342*** | .424*** |

*$p < .05$;

**$p < .01$;

***$p < .001$.

efficacy belief in all social roles. In the social role, the employee respondents obtained high scores (M = 13.46; SD = 2.29). The results in the individual scales of Impulsivity indicate that in the scales of the lack of premeditation (M = 7.98; SD = 2.04) and lack of perseverance (M = 7.79; SD = 2.17), the subjects had average scores, in the scales of negative urgency (M = 13.63; SD = 2.15) and positive urgency (M = 13.15; SD = 2.17) above average, and high scores in the sensation seeking scale (14.83; SD = 2.47). The analysis of correlations, the results of which are shown in Table 2, indicated that work engagement correlates positively with all scales of self-efficacy at the level of $p < .001$. Self-efficacy as a worker correlates with engagement at a moderate level (r = .45; $p < .001$). At the same time, there were negative correlations of work engagement with most Impulsivity scales: the lack of premeditation (r = -.22; $p < .001$), lack of perseverance (r = -.27; $p < .001$), positive urgency (r = -.12; $p < .01$) and negative urgency (r = -.12; $p < .01$). Only the need for sensation scale showed no relationship with work engagement. The results also indicated significant correlations between Impulsivity and self-efficacy factors. Except for the need for sensation, these were all negative correlations. Higher Impulsivity was not conducive to having self-efficacy beliefs in all measured social roles. For the key worker role of the study, the correlation with the lack of premeditation reached an average level (r = -.43,; pi < .001), and a similar level was obtained for the lack of perseverance (r = .40; pi < .001). The remaining Impulsivity scales obtained correlations at the level of low but significant scores (see Table 2). These results indicate that people who score highly on various dimensions of self-efficacy also usually show higher work engagement. In contrast, when it comes to the relationship between work engagement and Impulsive personality, the result obtained is exactly the opposite. Respondents demonstrating higher work engagement are characterized by low levels of impulsivity intensity, which means that they control their positive and negative actions, are able to plan tasks and conscientiously perform them.

### Analysis of hierarchical linear regression equations

Due to the occurrence of significant correlations, a regression analysis was conducted, where the explained variable was work engagement and the explanatory variables were Impulsivity

**Table 3. Regression equation coefficients.**

| Model | Included Variables | B | SE | β | t | p | r$_{a(b,c)}$ |
|---|---|---|---|---|---|---|---|
| 1 | (constant) | 45.688 | 4.091 | | 11.167 | .000 | |
| | Lack of premeditation | -.267 | .279 | -.063 | -.956 | .340 | -.047 |
| | Lack of perseverance | -.901 | .253 | -.225 | -3.562 | .000 | -.173 |
| | Negative urgency | -.071 | .246 | -.018 | -.289 | .773 | -.014 |
| | Positive urgency | -.167 | .251 | -.042 | -.664 | .507 | -.032 |
| | Sensation seeking | -.041 | .182 | -.012 | -.222 | .824 | -.011 |
| 2 | (constant) | 13.522 | 5.708 | | 2.369 | .018 | |
| | Lack of premeditation | .154 | .262 | .036 | .587 | .558 | .026 |
| | Lack of perseverance | -.529 | .243 | -.132 | -2.176 | .030 | -.097 |
| | Negative urgency | .076 | .231 | .019 | .331 | .741 | .015 |
| | Positive urgency | .044 | .237 | .011 | .188 | .851 | .008 |
| | Sensation seeking | -.288 | .171 | -.082 | -1.680 | .094 | -.075 |
| | Self-efficacy student | -.147 | .311 | -.038 | -.475 | .635 | -.021 |
| | Self-efficacy worker | 1.635 | .267 | .432 | 6.126 | .000 | .274 |
| | Self-efficacy homemaker | .235 | .246 | .065 | .958 | .339 | .043 |
| | Self-efficacy leisurite | -.268 | .175 | -.084 | -1.531 | .126 | -.069 |
| | Self-efficacy citizen | .389 | .172 | .116 | 2.256 | .025 | .101 |

and Self-efficacy (see Table 3). A block regression analysis conducted showed that engagement was statistically and significantly explained by Impulsivity (F = 6.93; df = 5; p < .001), albeit only a small percentage of the variance of the work engagement variable was explained ($R^2_{Adj}$ = .07). The addition of another bundle of variables—self-efficacy in roles increased the level of explained variance to a level higher than 20% ($R^2_{Adj}$ = .21), which was a very significant change ($F_{change}$ = 15.04; df = 5.383; p < .001). This section may be divided by subheadings. It should provide a concise and precise description of the experimental results, their interpretation, as well as the experimental conclusions that can be drawn.

In Model 1, including the variable Impulsivity, the only variable statistically significantly associated with work engagement was the lack of perseverance (β = -.225; t = -3.56 p < .001; r$_{a(b,c)}$ = -.173). A negative relationship between the two variables was observed i.e. the greater the lack of perseverance the lower the work engagement. In model 2, including Impulsivity and Self-efficacy together as explanatory variables, statistically significant relationships were for engagement and lack of perseverance (β = -.132; t = -2.17; p < .001; r$_{a(b,c)}$ = -.097) and the self-efficacy variable in the role of worker (β = -.432; t = 6.13 p < .001; r$_{a(b,c)}$ = -.274), and in the role of citizen (β = -.116; t = 2.26 p = .025; r$_{a(b,c)}$ = -.101). In the case of the self-efficacy variable, the relationship was positive, namely the higher the level of self-efficacy, the higher the level of work engagement.

## Analysis of moderation

According to the theses of JD-R theory, requirements moderate the relationship between resources and engagement, therefore the third step was to test this relationship. Self-efficacy as a worker was treated as a resource that positively influenced work engagement and this relationship was moderated by the interactions of each Impulsivity scale ($R^2_{-chng}$ = .0079; F = 3.9072; p < .05). In order to verify this hypothesis, the Process Hayes Model 1 [70] was applied separately for each Impulsivity scale. Moderation was found to be significant only for

**Table 4. Moderation effect of positive urgency on self-efficacy and work engagement relation.**

| Factors | B | SE | t | p | LLCI | ULCI |
|---|---|---|---|---|---|---|
| Constant | 34.60 | 12.92 | 2.68 | .008 | 9.205 | 59.991 |
| Self-efficacy as worker | -.07 | .91 | -.08 | .936 | -1.855 | 1.709 |
| Positive urgency | -1.83 | .94 | -1.95 | .052 | -3.684 | .017 |
| Interaction | .13 | .07 | 1.98 | .049 | .001 | .264 |

positive urgency (B = .133; SE = .067; t = 1.98; p < .05; LLCI = .0007; ULCI = 0.2643). The results of the analysis are shown in Tables 4 and 5.

Individuals with different levels of positive urgency intensity were characterized by the varying strength of the relationship between self-efficacy and work engagement. It was possible to notice the phenomenon that the relationship of self-efficacy and work engagement was stronger in subjects scoring higher on the positive urgency scale (see Fig 1), which was consistent with the model presented by Baker and Demerouti [14] and Bakker, Demerouti & Senz [15].

## Discussion

The theoretical assumption of our study was that Impulsivity as dispositional characteristics of employees would be negatively related to work engagement. This assumption was confirmed for four of its components: positive and negative urgency, the lack of perseverance and the lack of premeditation (see: Table 2). Only the fifth component of impulsivity–sensation seeking, does not correlate with work engagement. This relationship is logical and can be interpreted in two ways. For employees who exhibit high levels of impulsivity, including traits such as negative and positive urgency, lack of premeditation, and low perseverance, a significant decrease in work engagement is evident. In contrast, a decrease in both positive and negative urgency is evident in those who are highly engaged in work, which is deliberate and planned, and is accompanied by attention to the quality of the tasks performed. These results are logical within the context of the professional work environment. From the perspective of both the quality of tasks performed and relationships with colleagues, strong positive or negative emotional reactions to stimuli are neither desirable nor socially expected. When employees act under the influence of negative affect (negative urgency), their sensory and cognitive resources are limited, which can lead to mistakes in their work tasks. Employees who respond to even minor negative stimuli with intense negative emotions are not viewed positively by their colleagues. Similarly, high positive urgency, where an employee experiences strong positive emotions, can be problematic. Such employees may act rashly and lack the ability to plan and anticipate the consequences of their actions. Work engagement requires the ability to plan, anticipate, and persevere. In contrast, impulsivity traits like lack of premeditation and perseverance can hinder work engagement. Employees who lack perseverance may find repetitive tasks boring and perform them only routine-like, while those faced with difficult tasks may lose motivation and eventually abandon them.

**Table 5. Conditional effects of the focal predictor at values of the moderator.**

| Moderator–Positive urgency | B | se | t | p | LLCI | ULCI |
|---|---|---|---|---|---|---|
| **-1.1497** | 1.52 | .196 | 7.73 | < .001 | 1.131 | 1.903 |
| **-0.1497** | 1.65 | .178 | 9.28 | < .001 | 1.300 | 1.999 |
| **2.8503** | 2.05 | .255 | 8.03 | < .001 | 1.546 | 2.548 |

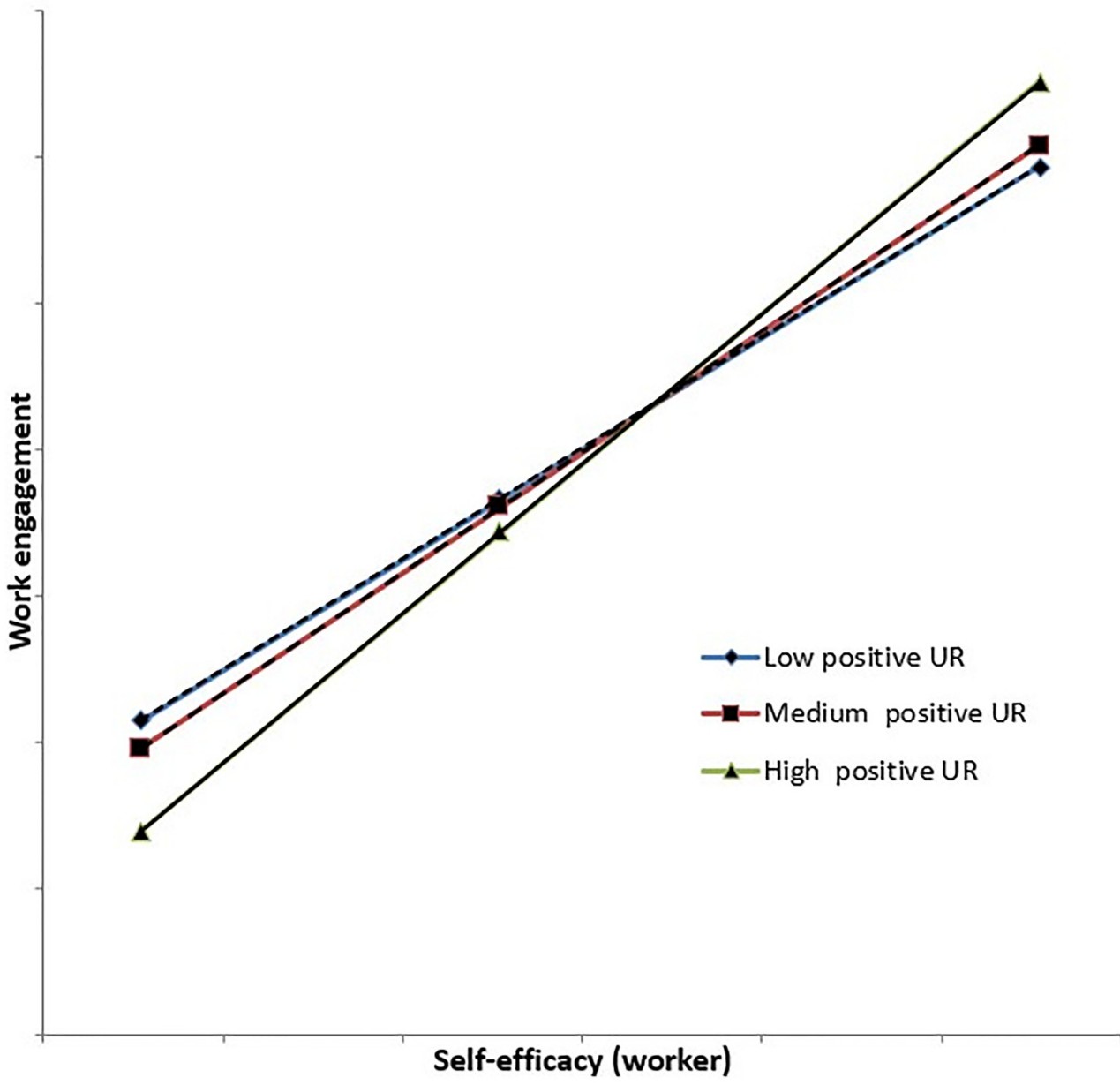

**Fig 1. Moderation effect of positive urgency on self-efficacy and work engagement relation.**

The lack of a significant association between work engagement and sensation seeking is consistent with expectations. Sensation seeking, as defined by Whiteside et al. and Lynam, involves seeking excitement and risky activities and being open to new and potentially dangerous experiences. While a work environment providing such stimulation is theoretically possible, it is quite rare in practice. Sensation seeking is inherently associated with risk, which may not align well with the typical requirements of most work environments. In general, the results obtained are in line with the majority of studies showing that these four dimensions of Impulsivity are positively related to a number of psychopathological behaviours, alcohol problems [51,57,58], eating disorders [52], and drug use [45,48]. Somewhat surprising is the result of a

positive relationship between work engagement and sensation seeking. This inconsistency is understandable when analyzing impulsive personality based on the Whiteside and Lynam theory. Firstly, several studies have confirmed that sensation seeking as measured by the UPPS-S questionnaire maintains relative independence from other dimensions of impulsive personality [71–73]. Secondly, research over the past decade has shown that sensation seeking correlates strongly with stress [48,64]. This result was interpreted to mean that high sensation-seeking reduces the strong need to receive gratification for completing a task, which lowers the feelings of social stress [74]. However, the cited interpretation refers to an analysis of the behaviour of rats bred, which the study was conducted on, so it has important limitations and requires robust verification in human studies. In our study, an influential factor, namely, social fear of Covid 19 coronavirus infection, may have been an outcome-modifying variable.

In the next step of the analysis, it was examined whether Impulsivity functions as a predictor of work engagement. In the J-DR model, demands are directly linked to the state of 'strain' [14] (p. 275) and only then negatively correlated with work engagement. The low level of explanation of the variance of work engagement by Impulsivity obtained in our study is in line with expectations from the model. A similar effect was also detected in other studies [18,74]. Only the lack of perseverance is a statistically significant factor when it comes to predicting work engagement (Tab.3). Hypothesis 1 was thus partially confirmed.

Additionally, the lack of perseverance was also found to be a negative predictor of work engagement. This is a logical and coherent relationship that indicates that low scores on the impulsivity component of the lack of perseverance are a predictor of work engagement. This means that people who are characterized by a very weak tendency to externalize expression in behaviour in the form of non-acceptance of emotional responses, lack of emotional clarity, lack of emotional awareness and acceptance of the lack of emotional responses show significant work engagement. This result of our study has also positive behavioural aspects that is internalizing behaviours, such as higher levels of self-control abilities. These regularities with respect to both externalization and internalization have been confirmed in several studies [75,76]. Another explanation for this predictive effect is also possible. The lack of perseverance on tasks is associated with unreliability in performance, disorderliness, susceptibility to boredom and quick discouragement. Several studies have found strong negative associations between the lack of perseverance and conscientiousness [77,78] and with the ability to maintain self-control [68]. Since some studies have detected that conscientiousness, understood as a Big Five personality trait, is strongly associated with perseverance [79,80] the result we obtained is consistent with other studies.

Based on the results obtained in other studies which analyzed Impulsivity in the UPPS model, it can be hypothesized that the lack of perseverance as a predictor of work engagement will also significantly reduce externalizing behaviours such as aggression [80,81] and violence [82]. The aim of our study, however, was broader than just uncovering the relationship between work engagement and Impulsivity traits. Our research based on JD-R theory also aimed to show the more complex role of Impulsivity in employee engagement by steering towards 'personal demands'.

It is noteworthy that in recent years, most studies considering the issue of work engagement have mainly looked for its associations in the area of important employee outcomes, such as contextual task performance, innovative work behaviour, proactive work behaviour, and reduced turnover intentions [83–86]. After the fairly well-identified work engagement construct, researchers are turning more towards specific in-work, engagement constructs, such as safety engagement, team engagement, pro-environmental engagement, engaging leadership and change engagement [87–90].

When research is focused on personal resources in the context of employee engagement, it has mainly considered factors such as resilience [91], self-efficacy [92], and psychological safety [93,94].

However, Impulsivity is by its nature something very different from these constructs. Impulsivity as a personal trait of an employee is still not quite seen in research in psychology as a serious limitation of its usefulness in an individual's work.

The associations between self-efficacy and work engagement are widely documented (see the letter from the theoretical section). However, these relationships are sensitive to the presence of other variables not previously considered.

This was demonstrated by our findings indicating that self-efficacy influences work engagement but that an important moderator of this relationship is Impulsivity. In particular, one of its dimensions–positive urgency (hypothesis 2) which moderates this relationship. Self-efficacy has a particularly strong impact on work engagement if the employee has one of the Impulsive personality traits–a high level of positive urgency. It is important to emphasize (as the results of our study showed) that the moderating effect of this relationship does not occur when the intensity level of the trait positive urgency is low. Self-efficacy has a particularly strong effect on work engagement if the employee has one of the Impulsive personality traits—a strong intensity of positive urgency. It is a state in which a person tends to act impulsively, but is accompanied by the experience of extremely strong positive emotions. In other words, a high level of the Impulsivity factor increases the strength of the relationship between self-efficacy and work engagement, in the same way as Bakker and Demerouti [14,15] describe the effects of demands in their model. This result is unexpected because we had assumed that impulsivity, as a personal trait, would constrain work engagement. The finding that one aspect of impulsivity—positive urgency—can influence work engagement, particularly when the employee also has high self-efficacy, suggests that impulsivity has a complex effect on quality of life at work. This indicates that impulsivity should be included in research on work engagement, as it is likely related to various variables not covered in our study. For instance, positive urgency might support the maintenance of good social relationships outside of work (such as with family, through social media, or within social circles). A strong balance between family and work can, in turn, enhance work engagement [95]. Recent research into the biological determinants of temperament, including impulsivity, confirms their complex impact on personal, social, and professional functioning. Positive affect, which is influenced by the configuration of the personality network for self-awareness, leads to creativity, pro-social behavior, and pro-social values [95], we believe it also contributes to greater work engagement. An important moderator of this network configuration is not only self-awareness but also self-efficacy.

JD-R theory [14,15] posits that there are two types of resources: personal and work-related. However, in the work published so far, there is no analogous division regarding demands.

The authors of the JD-R theory themselves indicate in the referenced article the prospect of extending the theory to include such a category. They cite perfectionism as a possible example of personal demands [14,15]. In our study we have dealt with Impulsivity, which is certainly a personal trait and not a work trait, so it must be considered as an element related to the 'personal' category.

The negative associations of Impulsivity with engagement detected in our study suggest, however, that it is not a resource but a factor that increases the psychological cost of work performance–a demand.

Although in the regression analysis only one dimension of Impulsivity was found to significantly influence engagement, while the direction of this relationship is confirmed by its placement in the theory as a personal demand.

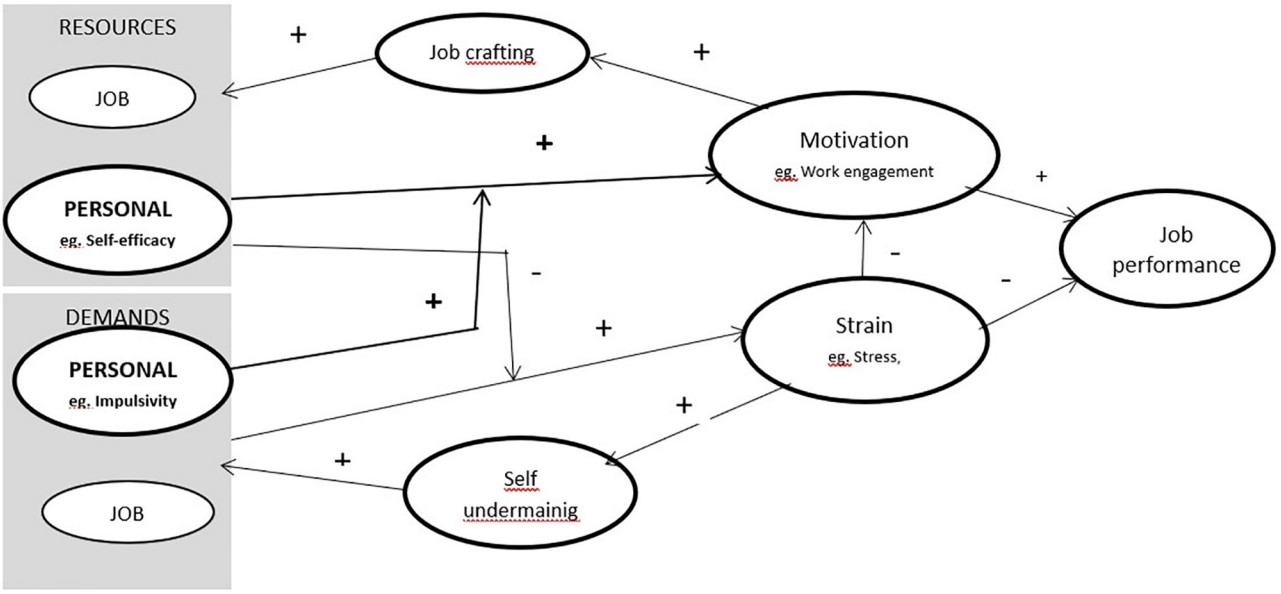

**Fig 2. Modyfied schemat of JD-R theory.**

We therefore propose to expand the schema described by Bakker and Demerouti [14,15] by adding one element. This need is also indicated by the negative feedback loops (self -undermining) mechanism, which should interact on personal rather than situational factors. Our proposal is shown in Fig 2.

The results of many studies on occupational burnout in the last three decades emphasize the fact of the complexity of this psychological phenomenon [96]. The lack of sufficient knowledge highlighting the differences between symptomatic and clinical (disease) conditions [97], confirms the need for further research. In the past two decades, the problem of burnout has very often been combined with the problem of employee work engagement [20,96–98].

On the other hand, the ongoing debate in psychology about whether job burnout and work engagement are related highlights that the problem is much more complex than supposed [99]. A good theoretical basis for both such research and multi-level analyses of these processes is offered by the JD-R theory.

The results of the research conducted in this model confirm that job burnout (stress) and work engagement are two distinct processes leading to different related outcomes [14,15].

Adopting the JD-R theoretical model therefore offers the opportunity to see the work process with its consequences in terms of both resources and work demands. In the research described here, particular emphasis is placed on the personal resource of self-efficacy, which in the JD-R model is qualified as a personal resource (personal resources) and Impulsivity as the individual disposition.

Although numerous studies indicate that self-efficacy positively influences work engagement [100–107], it was additionally detected in our study that this influence is stronger when the employee also possesses some Impulsivity traits (positive urgency).

The moderating effect of impulsivity on work engagement leads us to formulate a proposal to supplement the JD-R model in the area of the work demand category with a personality element (impulsivity, positive urgency), as illustrated in Fig 2.

Overall, we can conclude that the results of our research have shown that personal requirements should be added to the JDR theory in addition to personal demands. In this study, we

only included Impulsive Personality as defined by Whiteside and Lynam [63]. This dispositional factor is seen in psychological literature as a potential risk factor for psychopathological behaviour [108–111].

We know that Impulsive personality can lead to violent behaviours [112–115]. Research also shows that impulsivity and cannabis use are related to each other. Cannabis use increases impulsivity [116–118], and impulsivity increases the propensity for drug dependence, including cannabis [116,119–121]. A link has also been detected between cannabis use and psychosis among young people [122–125]; and their aggressive behaviour [112,113,124,125]. Psychosis and cannabis use, in turn, are high risk factors for serious violence [113,122,123,125]. Taking all these factors into account, it is important to emphasize that an Impulsive personality can be a serious factor in a person's poor quality of life. It causes many problems in social, personal and especially professional life. Our research has shown that it is also a serious moderator of work engagement. In the work environment, a decrease in work engagement can have negative consequences for the employee in the form of poorer employee evaluations and reduced pay. This in turn, will lead to the activation of negative behaviours in the work environment: aggression, alcohol and cannabis use. As Impulsive personality is an individual disposition that is very difficult to control on its own as these people require special care: interpersonal understanding, psychological support from management as well as colleagues, therapy and treatment. As the results of recent studies have shown, especially undertaking treatment at an early phase of cannabis use yields positive evolution of impulsivity [126].

Of course, in psychology, the concept of impulsive personality is understood in different ways, so further research on the impulsive personality of people functioning in the work environment is needed. Our research has also shown that the relationship of some impulsive personality traits (positive urgency and low levels of the lack of perseverance) with work engagement is multidimensional. The reasons for this multidimensional relationship may be individual, social as well as environmental. In future research, it would be interesting, for example, to investigate the relationship between impulsive personality and engagement when an employee performs tasks of varying degrees of difficulty, both short- and long-term, individually or in teams.

## Limitation

A certain limitation of our research is the fact that the people interviewed represented only certain sectors of the economy, trade, services, offices. The work demands they experience are therefore specific to this group of professions. The research results obtained should therefore only be related to these areas of work.

It is not excluded that for other professions the job demands will be different and then other Impulsivity traits may also prove to be significant moderators of work engagement such as the lack of perseverance or sensation seeking.

The period of the SARS-Cov-2 pandemic was associated with people's widespread fear of contracting the coronavirus. Therefore, it can be assumed that those doing on-site work experienced more severe stress than those working remotely.

Finally, the fact that the survey was conducted remotely may have affected potential response rates and thus the quality of the data obtained.

## Future research directions

Further analysis, particularly through longitudinal studies, is needed to establish causal relationships. Our findings suggest that while one dimension of impulsivity negatively affects engagement, another dimension moderates it. Future research should explore additional

dimensions of impulsivity that were not covered in this study. Moreover, further investigation is needed to explore the role of impulsivity in the context of work—particularly regarding engagement and burnout—across various populations, including different countries and specific occupations.

## Conclusion

We confirmed the key role of self-efficacy beliefs in enhancing work engagement. Our results show that the impulsivity factor known as "Lack of Perseverance" negatively correlates with work engagement, while the impulsivity dimension "Positive Urgency" moderates the relationship between resources and engagement. This highlights the complexity of impulsivity's impact on work engagement. Based on our findings, we support the inclusion of the "personal demands" element, which encompasses impulsivity, in the JD-R theory.

A practical conclusion of this research is to incorporate impulsivity assessments into the employee selection process for roles that require high commitment, such as leaders and managers. Specifically, attention should be given to candidates' "Lack of Perseverance."

A practical conclusion of this research is to incorporate impulsivity assessments into the employee selection process for roles that require high commitment, such as leaders and managers. Specifically, attention should be given to candidates' "Lack of Perseverance".

## Author Contributions

**Conceptualization:** Bohdan Rożnowski, Antoni Wontorczyk.

**Formal analysis:** Bohdan Rożnowski.

**Investigation:** Antoni Wontorczyk.

**Methodology:** Bohdan Rożnowski, Antoni Wontorczyk.

**Writing – original draft:** Bohdan Rożnowski, Antoni Wontorczyk.

**Writing – review & editing:** Bohdan Rożnowski, Antoni Wontorczyk.

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
