## [Decision Letter · Decision Letter 0]

17 Jan 2024

PONE-D-23-26784Impulsivity and work engagement: direct and indirect relationPLOS ONE

Dear Dr. Rożnowski,

Thank you for submitting your manuscript to PLOS ONE. After careful consideration, we feel that it has merit but does not fully meet PLOS ONE’s publication criteria as it currently stands. Therefore, we invite you to submit a revised version of the manuscript that addresses the points raised during the review process.

We look forward to receiving your revised manuscript.

Kind regards,

Heba E. El-Gazar

Academic Editor

PLOS ONE

Journal Requirements:

"This research received no external funding."

4. In the online submission form, you indicated that [Data available in the Institute of Applied Psychology Jagiellonian University, Department of Work and Organizational Psychology, on the request.]. 

6. We notice that your supplementary figures are uploaded with the file type 'Figure'. Please amend the file type to 'Supporting Information'. Please ensure that each Supporting Information file has a legend listed in the manuscript after the references list.

Additional Editor Comments:

Please review the comments carefully, as we cannot consider the study without thorough revision.

Reviewers' comments:

Reviewer's Responses to Questions

**Comments to the Author**

1. Is the manuscript technically sound, and do the data support the conclusions?

Reviewer #1: No

Reviewer #2: Yes

2. Has the statistical analysis been performed appropriately and rigorously? 

Reviewer #1: Yes

Reviewer #2: Yes

3. Have the authors made all data underlying the findings in their manuscript fully available?

Reviewer #1: Yes

Reviewer #2: No

4. Is the manuscript presented in an intelligible fashion and written in standard English?

Reviewer #1: No

Reviewer #2: Yes

5. Review Comments to the Author

**Dear authors,**

**I want to express my gratitude for allowing me the opportunity to review your research paper. It's evident that you've put considerable effort into your work, and I commend your dedication to advancing the field.**

**After a thorough review, I'd like to provide you with some feedback and recommendations to enhance the overall quality of your paper. I understand the importance of constructive feedback, and I believe these suggestions could strengthen your work and increase its chances of successful publication.**

**General Feedback**

**1. Clarity and Structure:**

Certain sections of your paper could benefit from improved clarity and concise language. Consider revisiting sentences and ensuring that the flow of ideas is seamless. This will enhance the overall readability of your paper.

**2. Grammar and Language:**

While the paper is generally well-written, there are a few grammatical issues that could be addressed to improve clarity. Be attentive to sentence structure and ensure consistency in verb tenses throughout the document.

**3. Additional Information:**

The paper appears to lack some essential elements that are typically expected in research articles. Including a more detailed literature review, a clear research question or hypothesis, and robust methodology would strengthen the overall framework of your study.

**I believe that incorporating these suggestions will significantly improve the quality of your research paper. I understand the effort and dedication required in academic writing, and I hope you find these recommendations helpful.**

**Detailed review**

**Abstract:**

*Clarity and Conciseness:*

The abstract is generally clear in presenting the study's purpose, methodology, and key findings.

However, certain sentences could be streamlined for better clarity and conciseness.

For example,

Original: "The study took up the idea of adding a new component to the Job Demands - Resources Theory (Bakker, Demerouti 12 & Sanz-Vergel, 2023), which should be called “personal demands”."

Revised: "The study proposes adding a new component to the Job Demands-Resources Theory, termed 'personal demands'."

Original: "As an example of personal demands, the construct of Impulsivity was studied."

Revised: "Impulsivity serves as an illustrative example of personal demands."

*Statistical Information:*

The abstract mentions a negative correlation between impulsivity and work engagement but does not provide effect sizes or statistical significance levels. Including such information would strengthen the credibility of the findings.

The citation for Bakker, Demerouti, and Sanz-Vergel (2023) should be removed from the abstract.

**Introduction: **

The introduction provides multiple perspectives on the definition of work engagement, particularly referring to Kahn's conceptualization. Consider consolidating these perspectives into a concise and cohesive definition for clarity.The discussion about work engagement being a temporary condition is important. Consider providing a brief example or illustration to make this concept more tangible for readers.The passage discusses stable resources and proposes the distinction of organizational and key personal resources. Consider briefly explaining these categories or providing examples to enhance understanding.Ensure consistency in terminology. For instance, the use of "work engagement" and "engagement at work" could be standardized for clarity.Carefully review grammar and syntax to ensure accurate expression of ideas. For example, in sentence 30, "in work engagement" might need clarification for smoother understanding.The authors effectively highlight the research gap related to personal demands within the JD-R model. The purpose of the study is clearly stated as investigating the impact of impulsivity on motivational states.The inclusion of prior research on impulsivity and its negative consequences, such as drug-taking, eating disorders, and risky behaviors, adds depth to the understanding of impulsivity. However, it might be helpful to briefly connect these findings to the workplace context to establish relevance.The second hypothesis introducing impulsivity as a moderator of the relationship between self-efficacy beliefs and work engagement is logically justified. This connection is consistent with the JD-R model and Bakker and Demeroouti's suggestion.

**Method: **

**
*Tools*
**

Ensure consistency in the use of terminology. For instance, the Utrecht Work Engagement Scale is mentioned in terms of "Vigour," "Absorbtion," and "Dedication." Consider using standardized terms such as "Vigor," "Absorption," and "Dedication" for consistency.Provide a brief overview of the five subscales of the SUPPS-P Impulsive Behavior Scale (positive UR, negative UR, premeditation, perseverance, and sensation seeking) for a better understanding of the impulsivity construct.

**
*Procedure*
**

The timeline for the survey is mentioned due to the COVID-19 pandemic, which is relevant. However, it might be beneficial to briefly explain why this specific time frame was chosen or if there were any external factors influencing the decision. The description of contacting potential respondents by phone and sharing the questionnaire through Google Forms due to the pandemic is clear. However, consider briefly explaining how this remote data collection method might influence data quality or participant responses.The informed consent process is described, which is good. However, explicitly mention that participants were informed about the study's purpose, conditions, and their rights. Ensure that the process aligns with ethical standards.Given the circumstances of the pandemic, the decision to avoid direct contact is justified. However, acknowledging this limitation and discussing any potential impact on response rates or data quality would add transparency.While it's mentioned that participants had the ability to withdraw at any stage, consider adding a sentence emphasizing that withdrawal would not result in any negative consequences.Ensure consistency in the presentation of dates. For example, "15-th" could be standardized as "15th" for clarity.

**
*Sample:*
**

The three provinces (Malopolskie, Podkarpackie, Swietokrzyskie) are mentioned, but it might be helpful to provide a brief context or rationale for choosing these specific regions. Are they representative of a larger population, or were they chosen for a particular reason?The distribution of education levels is detailed, but it might be beneficial to combine the categories for clarity. For example, the group "Bachelor of Arts" and "Master of Arts" together as "University Degrees" to simplify presentation.

**Results**

The detailed breakdown of impulsivity scales (lack of premeditation, lack of perseverance, negative urgency, positive urgency, and sensation seeking) is valuable. However, the use of terms like "average" and "above average" may benefit from more specific descriptors or comparisons.The correlation analysis is appropriately presented but consider providing a brief explanation or interpretation of what the correlations signify in terms of the relationships between variables. This can help readers understand the strength and direction of associations.Statistical information, including F-values, degrees of freedom, and p-values, is provided, which is essential. However, consider providing more detail on the coefficients and standard errors for each predictor variable to offer a comprehensive understanding of the results.It's mentioned that the Hayes Model 1 was applied for moderation analysis. In the section of statistical analysis (Missed), consider briefly explaining the key components of the Hayes Process Model 1 for readers unfamiliar with this statistical approach.

**Discussion:**

While the discussion provides an overview of the relationships between impulsivity and work engagement, consider a more detailed discussion of each component of impulsivity and its specific impact on work engagement.Ensure consistent use of terminology. For example, in some instances, the term "impulsivity" is used, while in others, it is referred to as "lack of perseverance." Maintain consistency in naming the components.For the significant findings, such as lack of perseverance being a negative predictor of work engagement, provide a more detailed interpretation of these results. Why is lack of perseverance negatively correlated with work engagement?If there are unexpected or contradictory findings, address them explicitly. For instance, why does sensation seeking correlate positively with work engagement while other components of impulsivity show negative correlations?When discussing the moderation effect of positive urgency on the relationship between self-efficacy and work engagement, provide a detailed explanation of the practical implications and potential reasons behind this moderation.Expand on the practical implications of the findings. How can organizations use this information to improve employee engagement or manage personal demands effectively?Please, conclude the discussion by summarizing the key findings, their theoretical and practical implications, and potential avenues for future research.

**Reviewer #2**: The manuscript titled "Impulsivity and work engagement: direct and indirect relation" focuses on incorporating a new component called "personal demands" into the Job Demands-Resources Theory (JDR theory), with a specific emphasis on impulsivity as a personal demand. The study aims to understand the relationship between impulsivity, self-efficacy, and work engagement among employees.

Novelty: The manuscript's novelty lies in its exploration of impulsivity within the JDR theory, an approach not extensively explored in existing literature. This adds a new dimension to understanding work engagement, especially considering the negative correlation of impulsivity with work engagement and its role as a moderator in the relationship between self-efficacy and work engagement.

Scientific Writing and Structure: The manuscript is well-structured, with clear sections including Introduction, Methods, Results, and Discussion. The writing is scientific and adequately references previous studies, providing a good context for the research question.

Methodology:

The research tools used are standardized and validated questionnaires, which strengthens the reliability of the data.

The survey was conducted through phone interviews, ensuring safety during the COVID-19 pandemic, and maintaining anonymity and voluntariness.

The sample size of 394 individuals is adequate, and the demographic details of the sample are well-documented.

But authors should report sample size or power

Analysis:

Descriptive statistics and correlations provide insights into the relationships between work engagement, self-efficacy, and impulsivity scales.

Why you used regression analysis and Moderation analysis?

Ethical Considerations:

The manuscript adheres to ethical guidelines for human subject research. However, it lacks specific details on the ethics committee's approval and the form of consent obtained from participants. Ethical Approval Details: Include specific details about the ethics committee's approval and the form of consent obtained.

Expand Discussion on Implications: Elaborate on the practical implications of the findings, particularly how understanding impulsivity as a personal demand can influence organizational strategies for enhancing work engagement.

Limitations and Future Research: Address any limitations more explicitly, such as the sample's geographic and sectoral representativeness, and suggest areas for future research.

6. PLOS authors have the option to publish the peer review history of their article (what does this mean?). If published, this will include your full peer review and any attached files.

Reviewer #1: **Yes: **Ahmed Hashem El-Monshed

Reviewer #2: **Yes: **Mohamed Zoromba

---

## [Author Response · Author response to Decision Letter 0]

7 Mar 2024

Dear Editor, 

We have carefully reviewed the comments received. Thank you, for careful reading of the text the comments made. We agree with them and have prepared a revised version of the article taking into account the comments made. In addition to substantive comments, the text has also been linguistically corrected by the proof-editor.

A detailed description of the changes made to the text in response to the reviews is included in the "Respons for Reviewers" file attached to the submission and bellow.

I hope that the corrections made meet your expectations.

Yours sincerely,

Response for Review

Reviewer:

The abstract is generally clear in presenting the study's purpose, methodology, and key findings. However, certain sentences could be streamlined for better clarity and conciseness. 

Thanks for the general positive opinion

Original: "The study took up the idea of adding a new component to the Job Demands - Resources Theory (Bakker, Demerouti 12 & Sanz-Vergel, 2023), which should be called “personal demands”.

We have edit text in line with the suggestion: The study proposes adding a new component to the Job Demands-Resources Theory, termed 'personal demands'.

Original: "As an example of personal demands, the construct of Impulsivity was studied." 

We have edit text in line with the suggestion: "Impulsivity serves as an illustrative example of personal demands."

Statistical Information: 

The abstract mentions a negative correlation between impulsivity and work engagement but does not provide effect sizes or statistical significance levels. Including such information would strengthen the credibility of the findings. 

We have include the coefficient and their statistical significance 

The citation for Bakker, Demerouti, and Sanz-Vergel (2023) should be removed from the abstract. 

We have removed it from the abstract

Introduction

The introduction provides multiple perspectives on the definition of work engagement, particularly referring to Kahn's conceptualization. Consider consolidating these perspectives into a concise and cohesive definition for clarity. 

We have clarified which frame we use as a basis for analysis (line 45)

The discussion about work engagement being a temporary condition is important. Consider providing a brief example or illustration to make this concept more tangible for readers. 

We added an example (lines 48-49)

The passage discusses stable resources and proposes the distinction of organizational and key personal resources. Consider briefly explaining these categories or providing examples to enhance understanding. 

We provided few examples of both kinds of resources from the cited publication of Bakker and DeVries.

Ensure consistency in terminology. For instance, the use of "work engagement" and "engagement at work" could be standardized for clarity. 

We standardized the key term as “work engagement”

Carefully review grammar and syntax to ensure accurate expression of ideas. For example, in sentence 30, "in work engagement" might need clarification for smoother understanding. 

The indicated deficiency has been corrected

The authors effectively highlight the research gap related to personal demands within the JD-R model. The purpose of the study is clearly stated as investigating the impact of impulsivity on motivational states. 

Thanks for positive opinion about our work

The inclusion of prior research on impulsivity and its negative consequences, such as drug-taking, eating disorders, and risky behaviors, adds depth to the understanding of impulsivity. However, it might be helpful to briefly connect these findings to the workplace context to establish relevance.

Thanks for this suggestion. We have expanded on the link between impulsive personality and work engagement

The second hypothesis introducing impulsivity as a moderator of the relationship between self-efficacy beliefs and work engagement is logically justified. This connection is consistent with the JD-R model and Bakker and Demeroouti's suggestion.

Thanks for positive opinion about our work

Tools 

Ensure consistency in the use of terminology. For instance, the Utrecht Work Engagement Scale is mentioned in terms of "Vigour," "Absorbtion," and "Dedication." Consider using standardized terms such as "Vigor," "Absorption," and "Dedication" for consistency. 

We standardized the key terms as "Vigor," "Absorption," and "Dedication"

Provide a brief overview of the five subscales of the SUPPS-P Impulsive Behavior Scale (positive UR, negative UR, premeditation, perseverance, and sensation seeking) for a better understanding of the impulsivity construct. 

Thanks for this suggestion. We have included sample questions of each scale.

Procedure 

The timeline for the survey is mentioned due to the COVID-19 pandemic, which is relevant. However, it might be beneficial to briefly explain why this specific time frame was chosen or if there were any external factors influencing the decision. 

Thanks for the pertinent comment. We have completed the clarification.The results presented in this text are part of a larger research project on personal requirements and resources related to work engagement. The choice of research timeframe was not influenced by external factors, but only by the realization of the empirical goal

The description of contacting potential respondents by phone and sharing the questionnaire through Google Forms due to the pandemic is clear. However, consider briefly explaining how this remote data collection method might influence data quality or participant responses. 

Thanks for this comment. The study was conducted during the SARS-COVID 19 pandemic, when a lock down was implemented in Poland and Europe. For these obvious reasons, the study had to be conducted remotely. Recommendations that all psychological research during this period be conducted remotely were issued by the Polish Psychological Association. Obviously, the period of the SARS-Cov-2 pandemic was associated with people's widespread fear of contracting the coronavirus. Therefore, it can be assumed that those doing on-site work experienced more severe stress than those working remotely. We have added a sentence in the limitations section indicating this potential limitation.

The informed consent process is described, which is good. However, explicitly mention that participants were informed about the study's purpose, conditions, and their rights. Ensure that the process aligns with ethical standards. 

Thanks for this comment. Of course, the research was conducted in accordance with the procedure of the Declaration of Helsinki after receiving a positive opinion from the Research Ethics Committee of the Institute of Applied Psychology of the Jagiellonian University. This clarification has been added to the text.

Given the circumstances of the pandemic, the decision to avoid direct contact is justified. However, acknowledging this limitation and discussing any potential impact on response rates or data quality would add transparency. 

Thanks for this suggestion. We added this sentence in section limitation

While it's mentioned that participants had the ability to withdraw at any stage, consider adding a sentence emphasizing that withdrawal would not result in any negative consequences. 

Thanks for this suggestion. We added this sentence

Ensure consistency in the presentation of dates. For example, "15-th" could be standardized as "15th" for clarity. 

We have standardized it as "15th"

Sample: 

The three provinces (Malopolskie, Podkarpackie, Swietokrzyskie) are mentioned, but it might be helpful to provide a brief context or rationale for choosing these specific regions. Are they representative of a larger population, or were they chosen for a particular reason? 

The coverage of three provinces provided the required cultural and environmental diversity in the research sample. The study of the others regions, in our opinion, was not necessary.

The distribution of education levels is detailed, but it might be beneficial to combine the categories for clarity. For example, the group "Bachelor of Arts" and "Master of Arts" together as "University Degrees" to simplify presentation. 

We have correct the data in Table 1.

Results 

The detailed breakdown of impulsivity scales (lack of premeditation, lack of perseverance, negative urgency, positive urgency, and sensation seeking) is valuable. However, the use of terms like "average" and "above average" may benefit from more specific descriptors or comparisons. 

The use of terms such as "average" and "above average" to describe the results obtained on various impulsivity scales is in line with the general standards adopted in psychology when analyzing personality traits related to impulsivity. Exactly the same terms were used by the authors of the Polish adaptation of the UPPS questionnaire. (Poprawa, 2019). Poprawa R, (2019). Research on the Polish short version of the Impulsive Behavior Scale UPPS-P. Alcoholism and Drug Addiction 32(1). DOI:10.5114/ain.2019.85767

The correlation analysis is appropriately presented but consider providing a brief explanation or interpretation of what the correlations signify in terms of the relationships between variables. This can help readers understand the strength and direction of associations. 

Thank you for this valuable remark. We overlooked it. We have added an explanation of this important connection in the text.

Statistical information, including F-values, degrees of freedom, and p-values, is provided, which is essential. However, consider providing more detail on the coefficients and standard errors for each predictor variable to offer a comprehensive understanding of the results. 

The coefficients and standard errors for each predictor variable are included in Table 2. We added more explanation of correlation to make clear our findings for readers.

It's mentioned that the Hayes Model 1 was applied for moderation analysis. In the section of statistical analysis (Missed), consider briefly explaining the key components of the Hayes Process Model 1 for readers unfamiliar with this statistical approach. 

A section on the overview of the statistics used was introduced, briefly explaining the main idea of moderation.

Discussion: 

While the discussion provides an overview of the relationships between impulsivity and work engagement, consider a more detailed discussion of each component of impulsivity and its specific impact on work engagement. 

Thank you for this valuable remark. Relationships between impulsivity and work engagement explained more clearly.

Ensure consistent use of terminology. For example, in some instances, the term "impulsivity" is used, while in others, it is referred to as "lack of perseverance." Maintain consistency in naming the components. 

Thanks for catching this inconsistency. We have corrected it

For the significant findings, such as lack of perseverance being a negative predictor of work engagement, provide a more detailed interpretation of these results. Why is lack of perseverance negatively correlated with work engagement? 

Thank you for this valuable remark. Relationships between impulsivity and work engagement explained more clearly.

If there are unexpected or contradictory findings, address them explicitly. For instance, why does sensation seeking correlate positively with work engagement while other components of impulsivity show negative correlations? 

Thank you for this valuable remark. Relationships between impulsivity and work engagement explained more clearly.

When discussing the moderation effect of positive urgency on the relationship between self-efficacy and work engagement, provide a detailed explanation of the practical implications and potential reasons behind this moderation. 

Thank you for this valuable remark. The moderation effect of positive urgency on the relationship between self-efficacy and work engagement explained more clearly.

Expand on the practical implications of the findings. How can organizations use this information to improve employee engagement or manage personal demands effectively? 

Thank you for this valuable comment. We have developed this issue further in the text.

Please, conclude the discussion by summarizing the key findings, their theoretical and practical implications, and potential avenues for future research 

Thank you for this valuable comment. We have developed this issue further in the text.

---

## [Decision Letter · Decision Letter 1]

24 Jul 2024

PONE-D-23-26784R1Impulsivity and work engagement: direct and indirect relationPLOS ONE

Dear Dr. Rożnowski,

Thank you for submitting your manuscript to PLOS ONE. After careful consideration, we feel that it has merit but does not fully meet PLOS ONE’s publication criteria as it currently stands. Therefore, we invite you to submit a revised version of the manuscript that addresses the points raised during the review process. 

I will make a decision based on the previous reviewer's reports and the current one. as well as my opinion revision, please address the comments of reviewer number 3, which are included in the letter 

editor's comments 

The title should clearly indicate the independent, dependent, and moderating variables, as well as the subject of the study. This will help the reader immediately grasp the key aspects of the research.

**Contextual Background:** The introduction would greatly benefit from a more comprehensive background on the JD-R theory and its evolution. This would provide a clearer understanding of the theoretical framework and how the current study contributes to it.

**Justification for Personal Demands:** A more detailed justification for introducing 'personal demands' within the JD-R framework would strengthen the theoretical foundation. This includes distinguishing personal demands from other constructs and explaining why impulsivity is crucial to study.

What about tool validity? Do the authors use tools in Polish language or English? A pilot study and Ethical considerations must be added to the text, especially if you collect data online; how did the authors secure that?

Nothing was mentioned about the sampling technique and sample size calculation. 

**Discussion:**

**In-Depth Analysis of Impulsivity Components:** The discussion could benefit from a more detailed analysis of each impulsivity component and its specific impact on work engagement. This includes explaining why certain components, like sensation seeking, correlate differently.**Contradictory Findings:** Addressing unexpected or contradictory findings in more detail and providing potential explanations would strengthen the discussion.**Practical Implications:** Expanding on the practical implications of the findings for organizational practices and interventions to manage impulsivity in the workplace would enhance the study's relevance.

**Conclusion:**

**Summary of Key Findings:** The conclusion could provide a succinct summary of the key findings, their theoretical contributions, and practical implications.**Future Research Directions:** Outlining potential avenues for future research, including longitudinal studies or experimental designs, would provide a strong closing to the paper.

**General Improvements:**

**Consistency in Terminology:** Ensuring consistent use of terminology throughout the paper, especially regarding constructs like 'work engagement' and its components.**Grammar and Syntax:** A thorough review of grammatical and syntactical errors to ensure clarity and professionalism in writing.Please submit your revised manuscript by Sep 07 2024 11:59PM. If you will need more time than this to complete your revisions, please reply to this message or contact the journal office at plosone@plos.org. Please include the following items when submitting your revised manuscript:
A rebuttal letter that responds to each point raised by the academic editor and reviewer(s). You should upload this letter as a separate file labeled 'Response to Reviewers'.A marked-up copy of your manuscript that highlights changes made to the original version. You should upload this as a separate file labeled 'Revised Manuscript with Track Changes'.An unmarked version of your revised paper without tracked changes. You should upload this as a separate file labeled 'Manuscript'.

We look forward to receiving your revised manuscript.

Kind regards,

Ayman Mohamed El-Ashry, Associate professor, Ph.D

Academic Editor

PLOS ONE

Additional Editor Comments:

I will make a decision based on the previous reviewer's reports and the current one. as well as my opinion revision, please address the comments of reviewer number 3, which are included in the letter

editor's comments

The title should clearly indicate the independent, dependent, and moderating variables, as well as the subject of the study. This will help the reader immediately grasp the key aspects of the research.

Contextual Background: The introduction would greatly benefit from a more comprehensive background on the JD-R theory and its evolution. This would provide a clearer understanding of the theoretical framework and how the current study contributes to it.

Justification for Personal Demands: A more detailed justification for introducing 'personal demands' within the JD-R framework would strengthen the theoretical foundation. This includes distinguishing personal demands from other constructs and explaining why impulsivity is crucial to study.

What about tool validity? Do the authors use tools in Polish language or English? A pilot study and Ethical considerations must be added to the text, especially if you collect data online; how did the authors secure that?

Nothing was mentioned about the sampling technique and sample size calculation.

Discussion:

In-Depth Analysis of Impulsivity Components: The discussion could benefit from a more detailed analysis of each impulsivity component and its specific impact on work engagement. This includes explaining why certain components, like sensation seeking, correlate differently.

Contradictory Findings: Addressing unexpected or contradictory findings in more detail and providing potential explanations would strengthen the discussion.

Practical Implications: Expanding on the practical implications of the findings for organizational practices and interventions to manage impulsivity in the workplace would enhance the study's relevance.

Conclusion:

Summary of Key Findings: The conclusion could provide a succinct summary of the key findings, their theoretical contributions, and practical implications.

Future Research Directions: Outlining potential avenues for future research, including longitudinal studies or experimental designs, would provide a strong closing to the paper.

General Improvements:

Consistency in Terminology: Ensuring consistent use of terminology throughout the paper, especially regarding constructs like 'work engagement' and its components.

Grammar and Syntax: A thorough review of grammatical and syntactical errors to ensure clarity and professionalism in writing.

Reviewers' comments:

Reviewer's Responses to Questions

**Comments to the Author**

1. If the authors have adequately addressed your comments raised in a previous round of review and you feel that this manuscript is now acceptable for publication, you may indicate that here to bypass the “Comments to the Author” section, enter your conflict of interest statement in the “Confidential to Editor” section, and submit your "Accept" recommendation.

Reviewer #3: (No Response)

2. Is the manuscript technically sound, and do the data support the conclusions?

Reviewer #3: Yes

3. Has the statistical analysis been performed appropriately and rigorously? 

Reviewer #3: Yes

4. Have the authors made all data underlying the findings in their manuscript fully available?

Reviewer #3: (No Response)

5. Is the manuscript presented in an intelligible fashion and written in standard English?

Reviewer #3: (No Response)

6. Review Comments to the Author

Reviewer #3: I read the manuscript entitled ”Impulsivity and work engagement: direct and indirect relations” with great interest. I believe that the authors presented an interesting research. I raised a few concerns while reading the manuscript, especially regarding how the literature gaps and corresponding contributions are presented. See all the comments below:

As a general observation, this study is a cross-sectional study, and it is incorrect to use causal terms in the description of the relations between variables. Please revise the manuscript and replace all the causal terms, including Discussion.

1. Please define the concept of personal resources.

2. Please argue why impulsivity could be considered a personal demand from the JD-R perspective. The authors should try to add the most important literature (2019-2023) to the topic, deeply demonstrate an understanding of the literature, review the literature critically, and point out limitations as well as conflicts. See Vîrgă, D., Schaufeli, W. B., Taris, T. W., van Beek, I., & Sulea, C. (2019). Attachment styles and employee performance: The mediating role of burnout. The Journal of Psychology, 153(4), 383–401.

3. Please separate the discussion section from Conclusion

4. The theoretical implication section needs to be created and completed. The research's theoretical implication needs to be discussed in relation to the first part of the article's conceptual framework.

5. I think the discussions could be enriched by focusing on these since it would better guide future research and alternative explanations for the findings.

6. What are the practical implications of this research?

7. PLOS authors have the option to publish the peer review history of their article (what does this mean?). If published, this will include your full peer review and any attached files.

Reviewer #3: No

---

## [Author Response · Author response to Decision Letter 1]

6 Sep 2024

Dear Editor,

Regarding the general comment about adjudicating relationships between variables, I want to note that I realize that cross-sectional studies do not provide a basis for inferring causal relationships. However, in the case of our studies , which analyze the relationships of variables, we assume the order of the variables in the relationship on the basis of JD-R theory. The values of correlation and regression coefficients confirm the existence of the relationship postulated by the theory and give information on the strength of the relationship. On the other hand, Impulsivity, which is considered in psychology to be a fixed trait determined genetically, cannot be the result of work engagement or belief of self-efficacy.

Reviewer Response

The title should clearly indicate the independent, dependent, and moderating variables, as well as the subject of the study. 

The title has been changed. Its first element is the explained variable. Then the explanatory variables and the group on which the study was done are listed.

The introduction would greatly benefit from a more comprehensive background on the JD-R theory and its evolution. 

The section on JD-R has been revised. The order of inclusion of each term in the theory and their definitions have been marked.

A more detailed justification for introducing 'personal demands' within the JD-R framework would strengthen the theoretical foundation. 

The definition of Personal demands has been added. Definitions of all categories of resources and requirements established on JD-R grounds have been placed close together.

What about tool validity? Do the authors use tools in Polish language or English? A pilot study and Ethical considerations must be added to the text, especially if you collect data online 

A Polish version of the scale was used in the study. All the scales used have their method validation publications in the Polish-language version. These publications are referenced in the text.

Nothing was mentioned about the sampling technique and sample size calculation 

The survey targeted the general working population. The sample size was due to obtaining sufficient test power and the requirements for multivariate regression methods. We were able to collect more people than the required criterion.

The discussion could benefit from a more detailed analysis of each impulsivity component and its specific impact on work engagement. This includes explaining why certain components, like sensation seeking, correlate differently 

Thank you for this comment. We have detailed each component of impulsivity to show how it correlates with work engagement.

Addressing unexpected or contradictory findings in more detail and providing potential explanations would strengthen the discussion 

Future Research Directions: Outlining potential avenues for future research, including longitudinal studies or experimental designs, would provide a strong closing to the paper 

In accordance with the reviewer's suggestions, we supplemented the discussion by also discussing unexpected and contradictory relationships obtained in the study

Summary of Key Findings: The conclusion could provide a succinct summary of the key findings, their theoretical contributions, and practical implications 

The indicated elements to the text have been added.

Consistency in Terminology: Ensuring consistent use of terminology throughout the paper, especially regarding constructs like 'work engagement' and its components.

Misspelled names of work engagement factors were identified and corrected.

Grammar and Syntax: A thorough review of grammatical and syntactical errors to ensure clarity and professionalism in writing. 

The text was subjected to further proof editing

Reviewr 3. 

1. Please define the concept of personal resources. 

The definition of personal resources and personal demands was added on p. 3.

2. Please argue why impulsivity could be considered a personal demand from the JD-R perspective. 

The authors should try to add the most important literature (2019-2023) to the topic, deeply demonstrate an understanding of the literature, review the literature critically, and point out limitations as well as conflicts. 

We also recognize some shortcomings of JD-R Theory. We point out the need to supplement the model with the construct of Impulsivity taking into account the comments from the article critical of JD-R. He wants to point out that we refer mainly to the latest version of the theory published by Bakker et al. 2023. (Bakker, A. B., Demerouti, E., & Sanz-Vergel, A. (2023). Job demands–resources theory: Ten years later. Annual Review of Organizational Psychology and Organizational Behavior, 10, 25–53. https://doi.org/10.1146/annurev-orgpsych-120920-053933)

3. Please separate the discussion section from Conclusion 

It has been separated by adding a separate section entitled Conclusion.

4. The theoretical implication section needs to be created and completed. The research's theoretical implication needs to be discussed in relation to the first part of the article's conceptual framework.

I think the discussions could be enriched by focusing on these since it would better guide future research and alternative explanations for the findings.

Thank you for this comment. The theoretical section on personal demands (Impulsivity) has been completed and explained in the discussion section.

6. What are the practical implications of this research? 

It was added in Conclusion.

---

## [Editor Report · Decision Letter 2]

25 Sep 2024

Work engagement, Impulsivity and, Self-efficacy among Polish workers. Moderating role of Impulsivity

PONE-D-23-26784R2

Dear respected authors

We’re pleased to inform you that your manuscript has been judged scientifically suitable for publication and will be formally accepted for publication once it meets all outstanding technical requirements.

Kind regards,

Ayman Mohamed El-Ashry, Associate professor, Ph.D

Academic Editor

PLOS ONE

Additional Editor Comments (optional):

the revisions detailed in the response to reviewers, the manuscript appears to be well-prepared for publication. The authors have addressed reviewers' comments, such as:

The title was modified to clearly state the independent, dependent, and moderating variables.

 The authors enhanced the theoretical background by providing more comprehensive explanations and justifications for introducing 'personal demands,' specifically impulsivity, into the JD-R framework.

They incorporated a detailed discussion of how different components of impulsivity, such as "lack of perseverance" and "positive urgency," affect work engagement, addressing the need for deeper analysis.

The authors conducted thorough statistical analyses, including regression and moderation analysis, to strengthen the validity of their conclusions.

 The authors also subjected the text to further proof editing for clarity and professionalism.

Reviewers' comments:

based on the previous revision and my evaluation  it seems that the authors addressed all the issue raised by the reviewers and the manuscript was ready for publication

---

## [Editor Report · Acceptance letter]

2 Oct 2024

PONE-D-23-26784R2 

PLOS ONE

Dear Dr. Rożnowski, 

I'm pleased to inform you that your manuscript has been deemed suitable for publication in PLOS ONE. Congratulations! Your manuscript is now being handed over to our production team.

Kind regards, 

on behalf of

Dr. Ayman Mohamed El-Ashry 

Academic Editor

PLOS ONE